# Imitation of a Pre-Designed Irregular 3D Yarn in Given Fabric Structures

**DOI:** 10.3390/polym14193992

**Published:** 2022-09-23

**Authors:** Tianyong Zheng, Wenli Yue, Xiaojiao Wang

**Affiliations:** School of Textile Science and Engineering, Tiangong University, Tianjin 300387, China

**Keywords:** fabric CAD, irregular 3D yarn, B-spline curve, fabric structure, key point mapping

## Abstract

The 3D CAD software has obvious advantages in appearance imitating and geometric structure modeling for fabrics. In contemporary 3D CAD fabric systems, only uniform yarns are involved in studies on fabric geometric structures, due to technological limitations, whereas objectives such as irregular/uneven 3D yarns have not been considered much. As the fabric structure or the central curve of the yarn changes, it is difficult to reflect the changed positions of the effect spots of the pre-designed uneven 3D yarns accordingly. In this paper, a key-point-mapping algorithm between the source yarn and the target curve is proposed to reflect the position change in effect spots when the fabric structure changes. By using the shape-preserving quasi-uniform cubic B-spline curve, a simple 3D irregular source yarn is designed using key points and setting their corresponding base cross-sections. The mapping is based on the principle that the lengths of the curve between the key points and the contours of the corresponding base cross-sections of the source yarn remain unchanged. Finally, the control grid of the new 3D yarn in the fabric structure is automatically generated. According to the examples and error analysis, the mapping technique can be applied to arbitrary given fabric structures, and the effect spots of the irregular 3D yarn are reasonably distributed as expected.

## 1. Introduction

The geometric structure of fabric is the extending and bending form of yarns in the three-dimensional space, which has a great influence on the appearance, physical properties and processing difficulty of fabric. Usually, the geometrical structure of fabric can be described in terms of two aspects: the cross-section of yarns and the central curve of yarns. The geometric structure affects almost all the properties of the fabric—appearance, strength, flexibility, drapability, porosity which is related to thermal insulation property, air ventilation, vapor permeability and electromagnetic performance. An accurate geometry model is used to imitate the appearance and pattern of the fabric, estimate the properties of the fabric, shorten the design and production cycle significantly and provide the basic data for apparel and industrial fabric design.

A computer-aided design (CAD) system for fabrics is widely used by designers to predict the appearance and geometrical structure of the fabric before weaving or knitting. After inputting the specifications of fabrics, the imitated image appears on the screen of the computer, which helps the designer to evaluate the design. Then, 2D graphical technology are normally used in the CAD systems, providing a quick and easy way to fulfill the target of imitation. Various types of yarns, uniform yarn or irregular yarns, are applied in design. When changing the fabric specification, CAD systems demonstrate the distribution of the effect spots of uneven fancy yarns on the fabric surface. However, the realism of imitation with 2D technology needs to improve dramatically as the edges of the yarns are blurred in the imitated image and the shading of the fabric is poor. There is another serious drawback that the 3D geometrical structure of the fabric is unavailable, which limits the application of the CAD system of 2D technology because the geometry of the fabric is necessary to predict its physical properties with the help of various finite software.

Therefore, 3D technology for modeling uniform yarns were developed to better construct the geometrical structure of fabrics in Liao [1], Lin [2], Lomov [3] and Sherburn [4]. The uniform 3D yarns are modeled either by facets, a Bezier surface or a B-spline surface, which are all determined by a control grid. However, a uniform/even yarn is only an ideal state, and most of the yarns are uneven in production and some yarns are purposely designed irregularly to form special effects on the fabric. For example, most fancy yarns designed irregularly.

To model irregular/uneven yarns, Jiang [5] and Gong [6] modeled the 3D yarn by setting the cross-section of the yarn as various super ellipses and ellipses, respectively. Software such as TexGen provide ways for modeling 3D yarn of various fixed cross-sections in the fabric structure, but fail to report the shapes of arbitrary cross-sections in a given fabric structure.

In Zheng [7], irregular 3D yarns are modeled by NURBS (Non-Uniform Rational B-Spline Curve). The shape-preserving quasi-uniform cubic B-spline curve is applied to fit the contour of the yarn path accurately, smoothly and stably. Two more adjacent points are inserted before and after an interpolating point, respectively, according to the direction of the all interpolating points, which ensures that the curve goes through all the interpolating points. Meanwhile, the quasi-uniform quadratic B-spline curve is applied to approximately to design the cross-section of a single yarn in the shape of the circle, ellipse, racetrack, lens, bowl, or round rectangle along the yarn path simultaneously. The shape of the cross-section is controlled by a 16-polygon formed by 18 control points. By manually assigning the specified base cross-sections at given positions, irregular 3D yarn effects appear in the structure of the woven fabric.

Actually, it is unreasonable to manually assign the contours of the cross-sections at given positions in a geometrical structure. If the contour of the pre-designed irregular yarn or the geometry of the fabric change, irregular 3D yarns bend in different forms and the positions of the effect spots change accordingly. The change in the distribution of the effect spots on the fabric surface has not been investigated. Indeed, such technological incompetence limits further in-depth application of the yarns, e.g., improved pattern design on apparels by Xue [8], where he developed a colored fancy yarn spun by three-channel digital ring spinning recently. The colored folded yarn comprises three irregular color yarn, which definitely enhances the esthetic effects of the apparels. Even in the CAD of 2D technology, where the effect spots can be viewed in the imitated image, the issue of the distribution of effect spots in the fabric structure is neglected or concealed by copying the part of the effected spots in the yarn image to the imitated fabric image directly.

The objective of this paper is to suggest/analyze a possible way of simulating the effect spot position distribution on irregular 3D yarn in any fabric structure. In other words, when the central curve of an uneven 3D yarn changes or the yarn itself moves arbitrarily, how will the effect spots in the pre-designed yarn change their positions accordingly given that no elongation and flatness occur?

## 2. Methodology

### 2.1. Choosing the Irregular Yarn

Due to the anisotropy of the irregular yarn, precise methods corresponding to different fancy yarn types are required. Slub yarn is the simplest type of irregular fancy yarn, in which slub knots are deliberately created to produce the desired effect. Therefore, an irregular yarn with a slub effect is a proper way to demonstrate how the effect spots distribute with the change in the fabric structure.

In most CAD systems for designing slub yarns, three parameters, namely, slub length, slub thickness and slub pitch, are used to control the contour of slub knots along the yarn (Figure 1); therefore, images of complicated slub yarns can be generated and put into a 2D fabric pattern. In order to avoid Moire effects, as a rule, slub yarns have a non-constant slub pitch between the slub spots.

In Li’s [9] study, these three control parameters were arranged freely along the central line of the uneven 3D yarn, and the yarns were only used to convert into 2D images to determine the evenness of yarns or Moire effects of the fabric. All the yarns were arranged in straight form, which could not reflect their real bent status in a fabric structure.

### 2.2. Representation of 3D Irregular Yarns

In this paper, 3D slub yarns are modeled by a shape-preserving quasi-uniform B-spline surface with OpenGL technology [10] as in Zheng [7]. Figure 2 illustrates the forming principle of the 3D yarn surface. The central curve of the yarn is represented by a shape-preserving quasi-uniform cubic B-spline curve (SPQUCBSC), which is determined by a series of key points as shown in Figure 2a. Each base cross-section corresponding to the key point of the yarn is characterized by a quasi-uniform quadratic B-spline curve, which is also defined by a series of control points in Figure 2c. The cross-section could be of an ellipse, or other shapes such as a racetrack or lens. The base cross-section *C*_1_ paralleling to *YOZ* plane in Figure 2a is used to calculate its shape conveniently. Then, all the base cross-sections are rotated to be perpendicular to the central curve to form cross-sections such as *C*_2_. The control points of each cross-section are rotated accordingly, creating the final B-spline control grid that determines the yarn shape as shown in Figure 2b.

A B-spline curve is determined by a series of control points, but an ordinary uniform B-spline curve does not go through the control points. From a designer’s perspective, the curve is expected to go through all the key points or interpolating points to ensure its shape. In order to meet the requirement, the inverse calculation is generally adopted, which is easily disturbed by the fluctuation of boundary conditions, however. To solve the problem, a robust algorithm was proposed in Zheng [7] that inserting additional control points would ensure the curve goes through the key points. Figure 3a shows an uneven yarn that goes through the given key points. The red dots are the key points controlling the central curve of the yarn as shown in Figure 3b. Two blue dots are automatically generated and inserted before and behind each red point (two end key points are excluded), as shown in Figure 3b, respectively, and each red key point locates at the midpoint of the segment line connecting the adjacent blue dots. Figure 3c shows the cross-sections of the yarn corresponding to all key points, and the sections are perpendicular to the center curve. Figure 3d shows the yarn control grid formed by connecting the control points of all cross-sections. To produce a uniform yarn, all the base cross-sections should be kept unchanged. If the base cross-sections are different in shape or size, an uneven 3D yarn with a slub effect will be modeled. When there are some larger cross-sections in a short distance, and then a slub knot will be designed. Therefore, each slub knot presents two aspects: the center of the cross-section and the shape of the cross-section. If a long slub knot is expected, and then there will be a long distance between the two key points where the two corresponding cross-sections are both larger than the normal one in size.

Usually, it is sufficient to design a straight slub yarn in a 2D CAD system. However, for universal applications, a curved irregular 3D yarn is created in this paper. The data of the control points of the cross-sections and the central line of the uneven 3D yarn as shown in Figure 3 or Figure 4a are listed in the Section A.1. The file format was illustrated in Zheng [11]. According to line 3 of the data, a 29 tex irregular yarn with a radius of approximately 0.069 mm is created and its shape is determined by 13 elliptical cross-sections. The following lines indicate the information of each cross-section, including the XYZ coordinates of the centers, types of the cross-sections and the related parameters. From the data, two cross-sections (Index No 5, and No 9, indexed from No 0) are enlarged by 1.7- and 1.4-fold, respectively. Therefore, the effect spots are formed at cross-section No 5 and No 9. It should be noted that there is only one key point controlling the location of the slub effect spot for each cross-section, so the length of the slub is not considered in the following description. Therefore, the slub length of the effect spot may change in the following examples although its relative position is fixed.

### 2.3. Description of the Geometrical Structure of Fabric

The geometrical structure of fabric is defined as the spatial status of the constitute threads. If the central curve of each constituent thread is correctly described, and then the 3D fabric structure is modelled. In this paper, the geometric structure of fabric is given by either calculating or measuring results. The central curve of a single thread in fabric is described as target curve which is controlled by a series of key points with X coordinates, Y coordinates and Z coordinates. A good example is shown in the Section A.2, where 10 central points of cross-sections are set to define the central curve of the single yarn in a given geometric structure repeat unit. As regards to the contour of the cross-sections, there is an assumption that no elongation and flatness occur when the yarn changes its path as this paper purely deals with a geometrical model rather than a physical model of the fabric structure. Therefore, the cross-sections of the source yarn will be copied directly to the yarns in fabric so that the effect spots of the source yarn will be kept.

### 2.4. Mathematical Expression of the Problem

To reflect an irregular/uneven 3D yarn in the fabric structure, it is necessary to investigate how the effect spots change their spatial positions when the central curve of the yarn changes freely. Figure 4a shows a 3D slub yarn with two knot effects in a repeat unit. If the central curve of the yarn changes from Figure 4b to Figure 4c, the change in the positions of the knots is shown in Figure 4d. In this converting procedure, the control points for the central curve of the new yarn are shown in Figure 4e.

From a mathematical point of view, the process can be described as the following sentences. Surface *Y* (the actual shape of the pre-designed source yarn) and surface *F* (the actual configuration of yarn in the fabric structure) are both modelled by a cubic B-spline surface. Both the central curves of the two surfaces are of SPQUCBSCs, and the base cross-sections are modelled by a quasi-uniform quadric B-spline curve. The key points sequence for central curve *P* of surface *Y* are given by the design process, and then the control points for the corresponding base cross-sections can be calculated accordingly. The sequence of key control points for the central curve *D* of surface *F* is also given by calculating or measuring the fabric structure.

To model a 3D yarn based on target curve *D* with OpenGL, the cross-sections should be calculated and assigned along curve *D* at the effect spots determined by key points at central curve *P* and the key points to determine the base cross-sections of the curve. Therefore, the key to the solution is to calculate the coordinates of the control points for the corresponding cross-sections on surface *F* according to surface *Y* of the source yarn.

Generally, an arbitrary point *D_s_* on the central curve *D* of surface *F* is set as the mapping start, and its counterpart point *P_s_* is also set arbitrarily on central curve *P* of surface *Y*. It is necessary to find the spatial coordinates of a random point *D_k_* (to be located) with a given distance of *L* behind the point *D_s_* along curve *D* and the spatial coordinates of the control points of the corresponding cross-sections on curve *D*.

To sum up, the idea to solve the change in the yarn central curve is to find the mapping relationship between the corresponding points and the cross-section from any given point on central curve *P* of surface *Y* to the central curve *D* of surface *F* when a pair of corresponding mapping starting points *P_s_* on curve *P* and *D_s_* on curve *D* are set.

### 2.5. Principle and Steps of Mapping

#### 2.5.1. Principle of Mapping

Key point mapping is used to solve the problem proposed in this paper. The method of mapping is to insert the key points from central curve *P* and the corresponding base cross-sections for surface *Y* to target curve *D* within a certain range of mapping length, and then combine them together to form a control grid of the new yarn. In this new control grid, the centers of the cross-sections are all on the target curve. Before mapping, two conditions have to be designated: (1) the positions on curve *D* for inserting the key points of curve *P* and (2) the corresponding base cross-sections at the original key points of curve *D*, which is why this process is named key point mapping. According to the principle that the length of the curve between the adjacent key points along source yarn *Y* remains unchanged, all the key points of curve *P* are inserted onto curve *D*, and generate a new curve, *D*^1^, which has the same shape of the original target curve, *D*. In practice, curve *D*^1^ is generated by copying curve *D* firstly. All the base cross-sections of surface *Y* are used to assign the cross-sections onto surface *F*. At each original key point on curve *D*^1^, the corresponding base cross-section is automatically generated in proportion to the curve distance of the key points with the known sections just added before and after on curve *D*^1^. If a pair of corresponding points on curve *P* and *D* for the mapping are set at different positions, the proportion will change and the contour of the new yarn will also change.

#### 2.5.2. Mapping Steps

The key-point-mapping processes are illustrated in the following diagrams. In Figure 5, we assume that central curve *P* of the source yarn is controlled by the sequence of key points {*P*_0_, *P*_1_, …, *P*_6_}, which are also the centers of the base cross-sections. In Figure 5, target curve *D* is controlled by the sequence of key points {*D*_0_, *D*_1_, …, *D*_4_}, which actually describes the geometric path of the yarn in the fabric structure. The change in curve *D* means the change in the geometry of the fabric. According to the algorithm of modelling the SPQUCBSC, curve lengths between every two neighboring key points along curve *P* are {*L*_0_, *L*_1_, …, *L*_5_} as shown in Figure 5a. For curve *D*, the curve lengths between every two neighboring key points are {*K*_0_, *K*_1_, …, *K*_3_} as shown in Figure 5b. In order to ensure that the new yarn has a path following that of target curve *D* and maintains the contour of the cross-sections as in surface *Y*, the key points of curve *P* and curve *D* are combined to form a new central curve *D*^1^ as the central path of the new yarn. The simplest mapping method is that *P*_0_ on the point curve *P* corresponds to point *D*_0_ on curve *D*. The key mapping steps are described as follows:

(1) If central curve *P* is straightened, the distance between the key points will be {*L*_0_, *L*_1_, …, *L*_5_}. Take the starting point as the coordinate origin, and obtain *X* coordinates of each corresponding key point along the *X* axis, as shown in Figure 6a.

(2) Similarly, straighten the central curve *D* and the lengths between each 2 adjacent key points along central curve *D* are {*K*_0_, *K*_1_, …, *K*_3_}. Take the starting point as the coordinate origin, and obtain *X* coordinates of each corresponding key point along the *X* axis, as shown in Figure 6b.

(3) According to the curve length between 2 adjacent key points along the source yarn *P*, the key point of the source yarn *P_i_* (*i* = 0, 1, …, 5, and *i* is an integer number in this paper) corresponds to point *P_i_^1^* on the new curve *D*^1^. Meanwhile, the contour of the corresponding base cross-section at *P_i_* on curve *D* is obtained and assigned to the base cross-section at *P_i_*^1^ on curve *D*^1^.

(4) Calculate the distance *L_m_* and *L_n_* between the key point *D_e_* (*e* can be any integer less than the number of the key points, 0 < *e* < 4 for this example) on curve *D*^1^ and the newly inserted nearest key points *P_M_*^1^ and *P_M_*_+1_^1^ on each side. Calculate all control points of the base cross-section corresponding to *D_e_* according to the distance proportion.

(5) According to *X* coordinates by straightening the curve (distance *L_i_* from the key point to the starting point on the source yarn *P*, or *K_i_* from the key point on target curve *D* to the mapping starting point), all the key points on curve *P* and curve *D* are mixed orderly in sequence to form the new central curve *D*^1^. Meanwhile, the corresponding base cross-sections are also arranged according to the order of these key points as shown in Figure 6c.

(6) Rearrange the sequence of key points similar to curve *D*. According to the distance of key point *P_i_* after an arbitrary key point *D_k_*, the order of sections of SPQUCBSC and parameter *t* are obtained for *P_i_*^2^, which is the corresponding point on curve *D* to key point *P_i_*^1^. Then, the *XYZ* coordinates of key point *P_i_*^2^ are calculated, and the *XYZ* coordinates of the control points for the corresponding base cross-sections are revised again by proportion. Based on the cross-section order determined in Step (5), a B-spline surface control mesh centered on curve *D*^1^ is formed to construct the uneven 3D slub yarn. Figure 7 is the schematic diagram showing the positions and sequence of key points on the newly generated curve.

Therefore, the algorithm for free conversion of the central curve of an irregular 3D yarn is as follows. (1) Calculate the distances between the key points along the central curve, which is the basis for the next three steps. (2) Find the counterpart of the key points of the source yarn on the target curve and set the corresponding cross-sections. (3) Set the base cross-sections at the original key points of the target curve according to the proportional interpolation method. (4) Combine and sort all the base cross-sections reflecting the spatial state of the uneven yarn in the order of the distance to the mapping start.

#### 2.5.3. Mode of Mapping

(1)Mapping from Origins

Figure 7 shows the simplest mapping—both the source yarn and the target curve start from their first key points. *P_s_*, the starting point to be mapped on the source yarn, just happened to be *P*_0_, the first key point of curve *P*, while *D_s_*, the starting point corresponding to *P_s_*, is also the first key point, *D*_0_, of target curve *D*.

(2)Mapping from Random Points

In actual fabric design and manufacturing, in order to avoid Moire effects in a large area in warp or weft directions and forming defects, the mapping starting point of warp or weft yarn should be randomly changed purposely. When considering the fabric width, the starting point of the source yarn and the target curve must be different at two adjacent weft yarns in continuous weft picking. Therefore, the starting point of the mapping of the source yarn, *P_s_*, is generally not the first key point of the source yarn, or *P*_0_; and the starting point of the target mapping curve, *D_s_*, is not its first key point, *D*_0_, either, as shown in Figure 8a,b, respectively. Let us suppose that the distance from *P_s_* to *P*_0_ is *L_s_*, and the distance from *D_s_* to *D*_0_ is *K_s_*. In Figure 8c, although *L_s_* < *L*_0_ and *K_s_* < *K*_0_, actually *L_s_* and *K_s_* are arbitrary values without any restriction. In calculating *P_i_*^2^, the key point on the mapped curve, the pair of the starting point *P_s_* and *D_s_* are both located at the distance *L_s_*–*K_s_* from *D*_0_ on the mapped curve, as shown in Figure 8c and Figure 9, respectively.

(3)Mapping over an Arbitrary Length

Figure 7 and Figure 9 show the mapping of the target curve for a single cycle. If the length of the mapped curve is not the length of a target curve repeat, but a random length, it makes the mapping more flexible and more widely used in practice as shown in Figure 10, where the curve length of the mapping is more than one cycle.

By designing the cumulative mapping length (the distance from the mapping starting position *D_s_*), the length of this mapping process and other parameters, multiple loop mapping can be achieved. If the cumulative length is very long, or the length from the starting point *D_s_* to *D*_0_ of the target curve is very large, there must be an appropriate way to locate *D_s_*. Therefore, it is necessary to find a way to represent the position of any point on the curve.

## 3. Algorithm

### 3.1. Sections of SPQUCBSC

A cubic B-spline curve is defined by a series of control points, and every four consecutive control points determine the expression of a curve section. Therefore, a cubic B-spline curve comprises cubic curve segments of different expressions. If the designed curve is expected to interpolate a serial of the key points *P*_0_, *P*_1_, …, *P_n_*, the control points of SPQUCBSC are *S*_0_, *S*_1_, …, *S*_3_×*_n_*, the number of the control points is *3* × *n +* 1, the curve segments in SPQUCBSC are *L*_0_, *L*_1_, …, *L*_3×*n*−3_, and the number of curve segments is 3 × *n* − 2. It should be mentioned that point *P_i_* is coincidental with point *S_i_*_×3_ (0 ≤ *i ≤ n*, and *i* is an integer number).

Figure 11 shows the curve interpolating five red key points, i.e., *P*_0_, *P*_1_, *…*, *P*_4_. To form such a shape-preserving curve, a sequence of 13 (5 × 3 − 2 = 13) control points {*S*_0_, *S*_1_, …, *S*_12_} is required. The curve comprises 10 (13 − 3 = 10) curve segments, expressed as {*L*_0_, *L*_1_, …, *L*_9_}. Each curve segment *L_i_* (*i =* 0, 1, 2, …, 9) is a cubic polynomial with different multinomial coefficients. These 10 curve segments form a complete quasi-uniform cubic B-spline curve.

Between the first two key points and last two key points, there are only two curve segments of the cubic curve, respectively. In contrast, there are three curve segments of the cubic curve between the middle key points. Therefore, for these middle key points, key point *P_i_* is the starting point of the curve segment indexed as *L*_3**i*−1_ (0 *< i < n*). In the example shown in Figure 11, *P*_1_ is the starting point of the curve segment indexed as *L*_2_ (2 = 3 × 1 − 1). *P*_2_ is the starting point of the curve segment indexed as *L*_5_ (5 = 3 × 2 − 1), while *P*_3_ is the starting point of the curve segment indexed as *L*_8_ (8 = 3 × 3 − 1).

### 3.2. Calculating the Distance between the Neighbouring Key Points

Defined by a series of control points {*S*_0_, *S*_1_, *S*_2_, *…*, *S_n_*}, the point *S_i_*(*t*) on the *i*th curve section of the B-spline curve defined by {*S_i_*, *S_i_*_+1_, *S_i_*_+2_, *S_i_*_+3_} is given by Formula (1) in Pigel [12]:(1)$$S_{i}\left(t\right)=\left[\begin{array}{cccc} 1 & t & t^{2} & t^{3} \end{array}\right]M_{3}\left[\begin{array}{c} S_{i}\\ S_{i+1}\\ S_{i+2}\\ S_{i+3} \end{array}\right]\begin{array}{cccc} & & & 0\leq t\leq 1\begin{array}{cc} & i=0,1,\ldots ,n-3 \end{array} \end{array}$$

The transformation matrix *M*_3_ is determined by *i* and *n* according to Zheng [7]. *t* is the parameter to control the position of the point on the curve. The formula can also be written as the following parametric Equation (2).
(2)$$\{\begin{array}{c} x=f\left(t\right)=a_{1}+b_{1}t+c_{1}t^{2}+d_{1}t^{3}\\ y=g\left(t\right)=a_{2}+b_{2}t+c_{2}t^{2}+d_{2}t^{3}\;\;\;\;\;\left(0\leq t\leq 1\right)\\ z=h\left(t\right)=a_{3}+b_{3}t+c_{3}t^{2}+d_{3}t^{3} \end{array}$$

Here, *x*, *y* and *z* are the coordinates of the point on the curve at *t*. *a_i_*, *b_i_*, *c_i_* and *d_i_* (*I* = 1, 2, 3) are determined by matrix *M*_3_.

In order to calculate the length of this curve segment *L_i_*, the arc differentiate length *ds* is given by Equation (3):(3)$$ds=\sqrt{{\left(dx\right)}^{2}+{\left(dy\right)}^{2}+{\left(dz\right)}^{2}}=\sqrt{{f\prime }^{2}\left(t\right){\left(dt\right)}^{2}+{g\prime }^{2}\left(t\right){\left(dt\right)}^{2}+{h\prime }^{2}\left(t\right){\left(dt\right)}^{2}}=\sqrt{{f\prime }^{2}\left(t\right)+{g\prime }^{2}\left(t\right)+{h\prime }^{2}\left(t\right)}dt$$

Then, the length of the curve segment *s* is calculated by Equation (4):(4)$$s=\overset{1}{\underset{0}{\int }}\sqrt{{f\prime }^{2}\left(t\right)+{g\prime }^{2}\left(t\right)+{h\prime }^{2}\left(t\right)}dt=\overset{1}{\underset{0}{\int }}u\left(t\right)dt$$

Let function $u\left(t\right)=\sqrt{m\left(t\right)}\;$
(5)$$m\left(t\right)=b_{1}^{2}+b_{2}^{2}+b_{3}^{2}+4c_{1}^{2}t^{2}+4c_{2}^{2}t^{2}+4c_{3}^{2}t^{2}+9d_{1}^{2}t^{4}+9d_{2}^{2}t^{4}+9d_{3}^{2}t^{4}+4b_{1}c_{1}t+4b_{2}c_{2}t+4b_{3}c_{3}t+6b_{1}d_{1}t^{2}\;+6b_{2}d_{2}t^{2}+6b_{3}d_{3}t^{2}+12c_{1}d_{1}t^{3}+12c_{2}d_{2}t^{3}+12c_{3}d_{3}t^{3}\;=b_{1}^{2}+b_{2}^{2}+b_{3}^{2}+\left(4b_{1}c_{1}+4b_{2}c_{2}+4b_{3}c_{3}\right)t+\left(4c_{1}^{2}+4c_{2}^{2}+4c_{3}^{2}++6b_{1}d_{1}+6b_{2}d_{2}+6b_{3}d_{3}\right)t^{2}\;+\left(12c_{1}d_{1}+12c_{2}d_{2}+12c_{3}d_{3}\right)t^{3}+\left(9d_{1}^{2}+9d_{2}^{2}+9d_{3}^{2}\right)t^{4}$$

Obviously, an exact integral expression to calculate the length of the curve cannot be obtained. In this case, Composite Simpson’s Rule [13] is used to approximate the calculation. According to the uniform distribution of *t* value (0 ≤ *t* ≤ 1), this curve section is divided into *w* (must be an even integer number) subintervals to obtain *w*/2 sub-segments of the curve. The length of these sub-segments of the curve, *G_j_* (*j* = 1, 2, …, *w*/2) is calculated as:(6)$$G_{1}=\frac{1}{3n}\left(u_{0}+4u_{1}+u_{2}\right)$$
(7)$$G_{2}=\frac{1}{3n}\left(u_{2}+4u_{3}+u_{4}\right)$$
(8)$$G_{j}=\frac{1}{3n}\left(u_{j-2}+4u_{j-1}+u_{j}\right)$$
(9)$$G_{\frac{n}{2}}=\frac{1}{3n}\left(u_{n-2}+4u_{n-1}+u_{n}\right)$$
where *u_i_* (*I* = 0, 1, …, *n*) = $u\left(\frac{i}{n}\right)$. According to Equations (8) and (9), the length of the curve segment *L_i_* defined by control points *S_i_*, *S_i_*_+1_, *S_i_*_+2_, and *S_i_*_+3_ is approximately calculated as the sum of the sub-segments of the curve as shown in Equation (10).
(10)$$L_{i}=\overset{1}{\underset{0}{\int }}u\left(x\right)dx\approx \frac{1}{3n}\left[\left(u_{0}+u_{n}\right)+2\left(u_{2}+u_{4}+\ldots +u_{n+2}\right)+4\left(u_{1}+u_{3}+\ldots u_{n-1}\right)\right]$$

The larger the even number *w*, the more accurate the calculation.

### 3.3. The Representation of an Arbitrary Point on a SPQUCBSC

#### 3.3.1. Definition of Anterior and Posterior on a Curve

On the curve, there are two points *A* and *B*, respectively, and the distances between the two points to the starting point *S_s_* of the curve are *L_A_* and *L_B_*, respectively. If *L_A_* < *L_B_*, point *A* is said to be in front of point *B* at *L_B_*–*L_A_* and point *B* is behind point *A* at *L_B_–L_A_*. So, point *A* is the anterior and point *B* is the posterior.

#### 3.3.2. Representation of a Given Point on a SPQUCBSC

There are two methods to locate a given point on a SPQUCBSC.

(1)Index Number of the Curve Segment + Parameter *t*

According to definition (1) or (2) of a B-spline curve, the *XYZ* coordinates of any given point on the curve are accurately determined through four control points and parameter value *t*. Once four control points are known, the index number of the curve segment defined by them is determined. This representation is equivalent to the form of “the index number of the nearest shape-preserving anterior control point *A* + the distance behind point *A* along curve”. The method is mathematically easy to understand, but not intuitive in locating an arbitrary point on the curve.

(2)Index Number of the Anterior Key Point A + Distance to Point A along the Curve

The second way to define a given point on a SPQUCBSC is to use the mode of “index number of the anterior key point *A* + distance behind point *A* along the curve”, which is intuitive for the user to understand, but difficult to map. Therefore, it must be converted into the mode of “the index number of the nearest shape-preserving anterior control point *B* + the distance behind point *B* along curve”, and then converted into the mode of “the index number of the curve segment + parameter *t*”.

Based on the way of locating a given point, the point *A* shown in Figure 11 on curve *L*_6_ can be defined or located in either of the following ways: ① at the distance of *L* after key point *P*_1_, requiring *L*_2_ + *L*_3_ + *L*_4_ + *L*_5_ < *L* < *L*_2_ + *L*_3_ + *L*_4_ + *L*_5_ + *L*_6_; ② at the distance of *L_A_* after key point *P*_2_, demanding 0 < *L_A_* < *L*_6_; ③ on the curve segment indexed No 6 (or curve *L*_6_) defined by control points *S*_6_(*P*_2_), *S*_7_, *S*_8_ and *S*_9_, at parameter *t* = *t_A_*, where *t_A_* is to be calculated later.

### 3.4. Locating of the Corresponding Point on the Given SPQUCBSC

The key to this algorithm is to find the *XYZ* coordinates of a given point on a curve according to the index number of the anterior key point and the distance after the key point. It is actually to calculate the index number of the curve segment and parameter *t* of the point on SPQUCBSC. This algorithm can not only locate the starting point of the mapping, but also search the mapped point (corresponding point) on curve *D* from any given point on curve *P*.

The following description takes the searching of a given point on central curve *P* of the source yarn as an example to describe the steps of the algorithm.

(1)Calculation of the Repeat Unit Length of the Curve

The repeat unit length of the curve is the length of the curve that passes through all the key points, and is the sum of the lengths of the curve segments defined by the sequence of shape-preserving control points. The repeat unit length *R_s_* of curve *S* is calculated by Equation (11), respectively. Supposing the number of the key points to interpolate SPQUCBSC *S* is *n* + 1, and then the number of the curve segments is 3*n* − 2 (*or h =* 3*n* − 3).
(11)$$R_{s}=\sum_{i=0}^{h}L_{i}\;\;\;\;\;\;\;\;\;\;\;\;\left(h+1:numbr\;of\;the\;curve\;segments\;on\;curve\;S\right)$$

(2)Calculation of the Number of Repeats *N_s_*

When the mapping length behind a given starting point is greater than the repeat unit length of a curve, the number of repeats (*N_s_*) of mapping should be calculated. The calculation method is equal to the quotient of the sum of the distance *D_s_* of the point from mapping starting point and the mapping length *L* divided by the repeat unit length *R_s_* of the curve, and the integer part of the quotient is set as *N_s_.* The formula is shown in Equation (12). Be noted that it’s not rounded up or rounded down.
(12)$$N_{s}=\left(integer\right)\;\frac{D_{s}+L}{R_{s}}$$

Therefore, to the point far away from the mapping start, the coordinate differences resulting from the repeats (*N_s_*) should be added, which will be explained in step (5).

(3)Calculation of the Index Number of the Curve Segment of SPQUCBSC

According to the accumulated value of the length *L_i_* of each curve section, the index number (*N_x_*) of the shape-preserving curve section where the point is located is determined, which is actually the index number of the nearest anterior key point in front of the point to search. According to Equation (13), the residue mapping length *L_r_* that exceeds a number of complete repeats is calculated.
(13)$$\;L_{r}=L+Ds-N_{s}\cdot R_{s}$$

Then, solve the inequality
(14)$$\sum_{i=0}^{h}L_{i}<L_{r}$$

A series of *i* values that satisfy the condition are obtained. Among them, the maximum value of *i* is selected as the index number *N_x_* of the curve segment, or, *N_x_* = Max{*i*}.

(4)Calculation of Parameter *t*

Composite Simpson’s Rule dictates that when calculating curve segments, each curve segment is divided into *w* (an even integer number) subintervals. According to the method similar to calculating the index number of the curve segment, the index number *T* (*T* = 0, 1, …, *w*/2) of the subinterval of the curve segment containing the corresponding point is obtained. Thus, parameter *t* is determined.
(15)$$\mathrm{Let}\;Q_{r}=L_{r}-\sum_{i=0}^{N_{x}}L_{i}$$

Again, solve the inequality
(16)$$\sum_{i=0}^{N_{x}}G_{i}<Q_{r}$$

A series of *i* values that satisfy the condition are obtained and *T* is the maximum value of *i*, or *T* = Max{*i*}. Finally, parameter *t* is calculated by Equation (17).
*T* = (*float*) 2 × *T*/*w*(17)

Since *w* is set as 20 in this paper, *t* is thereby one of the 11 values from 0, 0.1, 0.2, 0.3, …, 1.0.

(5)Calculation of the *XYZ* Coordinates of the Mapped Point

After *N_s_* (number of repeat units), *N_x_* (index of the curve segment) and *t* are all calculated, the coordinates of the mapped point (*X*^1^, *Y*^1^, *Z*^1^) can be calculated according to Equation (2). Then, the final coordinates (*X_t_*, *Y_t_*, *Z_t_*) of the mapped point at parameter *t* are calculated by Equation (18):(18)$$\{\begin{array}{c} X_{t}=X^{1}+\left(N_{st}-N_{ss}\right)\cdot (X_{n}-X_{0})\\ Y_{t}=Y^{1}+\left(N_{st}-N_{ss}\right)\cdot (Y_{n}-Y_{0})\\ Z_{t}=Z^{1}+\left(N_{st}-N_{ss}\right)\cdot (Z_{n}-Z_{0}) \end{array}$$

Here, *N_st_* means the number of repeats *N_s_* at given point of parameter *t*, *N_ss_* is the number of repeats *N_s_* at the mapping starting point. *X_n_* is the *X* coordinate of the last point of original curve *S* while *X*_0_ is the *X* coordinate of the first point of the original curve *S*. The rule is also applied to *Y_n_*, *Y*_0_, *Z_n_* and *Z*_0_.

For searching the coordinates of any point on target curve *D*, the algorithm is exactly the same.

### 3.5. Calculation of the Control Points of the Base Cross-Section Generated Automatically

The base cross-section (*C*_1_, *C_a_*, *C_M_* and *C_B_*) as shown in Figure 12 is actually a curve in a 2D plane parallel to plane *YOZ* plane, and is determined by 18 control points. All the base cross-sections (such as *C*_1_) should automatically rotate to the plane that is perpendicular to the central curve of the yarn, i.e., *C*_2_, to form the final controlling mesh for the uneven 3D yarn.

The coordinates of all the 18 control points shown in Figure 2c for a cross-section are set relatively to its center. The newly generated base cross-section corresponding to any point between the two base cross-sections before and after which are at the known spacing is smoothly interpolated according to proportion of length, the method is demonstrated in Figure 12.

Assuming that the base cross-sections *C_a_* and *C_b_* corresponding, respectively, to point *S_a_* and *S_b_* at the central curve have been calculated, in other words, the coordinates of a pair of the corresponding control points *P_a_* and *P_b_* on the base cross-section are known, it is necessary to calculate the coordinates of the corresponding control point *P_m_* on the cross-section at the *S_m_* point on the central curve.

Here, we can re-write the distances between the point pairs, *S_a_* − *S_m_*, and *S_m_* − *S_b_*, which are all set along the curve, as *L_ma_* and *L_bm_*, respectively. *P_m_* can be obtained according to the proportional relationship by Equation (19):(19)$$\frac{P_{m}-P_{a}}{L_{ma}}=\frac{P_{b}-P_{m}}{L_{bm}}$$

Similarly, the coordinates of the other 17 control points on the base cross-section *C_m_* can be calculated automatically.

### 3.6. Sort and Combine All the Key Points and the Corresponding Base Cross-Sections

After calculating, all the corresponding points of the key points on the source yarn and the key points on target curve *D* will be combined and ascendingly sorted to form the mapped curve *D*^1^ according to their distances to the mapping starting point *D_s_*. Meanwhile, the base cross-sections corresponding to all these key points are also sorted in the same order. Thus, a control mesh of new yarn is formed and the uneven 3D yarn can be drawn by OpenGL.

The sorting process can be solved by the conventional bubbling algorithm, which will not be described here.

## 4. Results

### 4.1. Imitation of an Irregular Yarn in Free Space

The following examples show the mapping effect of different mapping length at different starting mapping points with the same source yarn and target curve. The specifications of the source yarn and the target curve to test are listed in Table 1. The length of the curve repeat is calculated by Composite Simpson’s Rule according to the data in Section A.1 and Section A.2.

By changing the mapping starting points of the source yarn, target curve and mapping length, four experiments are tested to demonstrate the mapping results. The mapping parameters are shown in Table 2 and the final effects are demonstrated in Figure 13. It should be noted that all the key points are indexed from No 0. The two corresponding mapping starting points of the source yarn and the target curve are both set by the mode “index number of the anterior key point A + distance behind point A along the curve”. In example No 1, the simplest mapping from origins is used and the resulting effect is shown in Figure 4d. In example No 2, the mapping starting point of the source yarn is not its first key point but its corresponding point is the first key point of the targe curve, and the mapped effect is shown in Figure 13a. In example No 3, the mapping starting point of the source yarn is the first key point but its corresponding point is not the first key point of the targe curve, and the mapped effect is shown in Figure 13b. In these three examples, the mapping length is a full repeat unit of the target curve, thereby a full knitted loop is shown in Figure 4d and Figure 13a,b, respectively, although the three loops looked quite different. In example No 4, there is displacement of mapping starting points from both the origins of the source yarn and the target curve, and the mapping length is more than a repeat unit, so more than one loop repeat is shown in Figure 13c. It is intuitively reliable that the key-point-mapping algorithm works well since the effect spots distribute on the target curve as expected.

Figure 14c shows another three different effects, respectively, by mapping the yarn to target curve in Figure 14a at different mapping starting points. Figure 15a shows the enlarged view of local image of Figure 14c, while Figure 15b shows the corresponding control points for the mapped yarn. Figure 15c shows the magnified mapped yarn if the target curve in Figure 14b is applied, and the control points are shown in Figure 15d. The data of the key points for target curve in Figure 14a and targe curve in Figure 14b are listed in Section A.3 and Section A.4, respectively. From the data, two curves are both 2D curves since all the *X* coordinates of the key points are constant. So, a real yarn is bent to imitate the target curve to verify the validity of the mapping. In Figure 16, the green dots on the yarn indicate the effect spots of the irregular yarn. Figure 16a shows the red dots on the real yarn are arranged as the XYZ coordinates in the file of Section A.2 while Figure 16b shows the source yarn be stretched straight. The smallest scale on graph paper is 0.1. According to scale, the length of the source is 2.7, which is approximate to the curve length calculated by the algorithm based on Composite Simpson’s Rule. The mark “×” in Figure 16c indicates the key points of the target curves defined in the Section A.3; meanwhile, three effect yarns with different mapping starts are bent and passing through the marks smoothly. Figure 16c demonstrates that the distribution of the green effect spots is quite similar to that in the simulated image of Figure 14c.

### 4.2. Applying Irregular 3D Yarns in Fabric Structures

Once the central curve of an irregular 3D yarn has been freely transformed, it can be applied to various industrial applications, such as electronic blackboard, 3D fabric structure modeling and appearance simulation for woven, knitted and braided fabrics. The source yarn in the Section A.1 is used in the following examples again.

#### 4.2.1. Electronic Blackboard

In simulating the electronic blackboard of uneven 3D yarn as shown in Figure 17, all the target curves are actually a series of parallel straight lines, and each line can be designed by only four key points. In mapping, the target curves (actually, straight lines) are processed from their first key points. However, the invisible yarn segments on the back of the blackboard should be considered, which can be achieved by setting different cumulative mapping lengths for each mapping.

#### 4.2.2. Imitation of Woven Geometric Structures

In order to simulate the geometric structure of woven fabrics, it is necessary to randomly distribute the mapping start of the central curve of the warp yarn in order to avoid Moire effects in large area. The mapping of weft yarns is similar to that of electronic blackboards in setting the accumulative mapped length, but different weft yarn has a different mapping setting depending on the direction of picking. If it is applied for a shuttleless loom, the weft yarn is picked from the same side. The mapping starting points of all weft yarns are set the same, and the cumulative length of each yarn should be increased by one fabric width based on the previous weft mapping. For a shuttle loom, if the weft yarn is picked from both sides, and then it may be necessary to reverse the direction of the key points of the target curve representing the original fabric structure. Figure 18 shows the geometrical structure of plain woven fabric by picking the 3D effect yarn from same side of the loom. In this example, the *XYZ* coordinates of each interlacing point are roughly calculated. Supposing it is a balanced plain structure, all the interlacing points are considered to be evenly distributed, and the distance between the warp thread and the weft thread is set as one diameter of yarn. The sequence of the interlacing points forms the target curve. In order to better control the path of the yarns in structure, a middle point is inserted between two neighboring interlacing points. The data file of woven structure or the central curves of all the yarns for Figure 18 is shown in the Section A.5. In each target curve repeat unit, there are nine key points for four interlacing points. In the file, the second data ‘12’ of the line 2 means that there are 12 target curves in the structure. According to the coordinates of the center of each cross-section, six ends and six picks in the geometric structure of a woven fabric are set in the file. Actually, one and half repeat of the target curve is mapped in the example as shown in Figure 18.

#### 4.2.3. Imitation of Weft-Knitted Geometric Structure

The geometric structure of the fabric is changed as shown in the Section A.6, which is roughly obtained for a plain knitted stitch. According to the data, there are 4 yarns in the knitted structure, each loop unit is controlled by 10 key points. The central curve of the yarn at the 1st wale is just the same as the target curve in the Section A.2, and the key points controlling the remaining three yarn in the next three wales are just obtained by translating the previous yarn upward for a fixed distance, respectively. More than one repeat of the target curve is mapped in the example, which means a large area of the fabric can be imitated. The mapping starting point on the target curve of the weft-knitted fabrics is similar to mapping the weft yarns in a shuttleless-woven fabric.

Figure 19 shows an imitated geometrical structure of the weft knitted fabric with the 3D uneven yarns. From the imitated image, the distribution of each effect spots is clearly demonstrated.

From the above examples, it can be seen that the effect knots in 3D slub yarns are distributed as expected at the reasonable positions of different fabric structures by using the key-point-mapping technique.

## 5. Discussion

When the shape of the target curve and the final mapped curve are carefully observed, some slight differences in shape will be noticed although curve *D* and curve *D*^1^ are very similar. In the following part, rationalization to such phenomena will be addressed.

### 5.1. Error Analysis

#### 5.1.1. Source of Error

The sources of difference are originated from three parts:

(1) According to the principle of mapping, the number of key points of target curve *D* is different from that of the final merged curve *D*^1^. As the key points of the source yarn are inserted, the number of control points on the central curve of the final yarn changes, and so does the expression of each curve segment. Therefore, the two curves are definitely different, and the error occurs.

(2) When calculating the coordinates of a given point on SPQUCBSC, the length of each curve segment needs to be calculated accurately. However, the length is calculated approximately by summation, which is not a deterministic value, and its accuracy depends on the number of the intervals *w* of each curve segment. The larger *w* is, the higher the accuracy and the smaller the error.

(3) When calculating the final spatial coordinates of points on the curve, parameter *t* needs to be determined. However, the value of *t* is a lookup, there is not a continuous change but a jump when *t* changes. The jump value is determined by the number of intervals *w* and skips by at least 2/*w* each time, resulting in deviation in the calculation of the coordinates of the given point. Obviously, the larger *w* is, the smaller the error.

#### 5.1.2. Margin of Error

To improve the calculating speed on curve length and save memory, the interval number *w* should remain within a certain range. In the examples in this paper, *w* = 20. Taking the source yarn in Figure 4a as an example (same position of all the key points except that two knots of the same thickness), three different target curves are mapped with four different lengths, respectively. Table 3 shows the accumulative length calculation error values under 12 different mapping conditions. Due to the huge amount of the data, the lengths between each effect spot have not been listed.

It can be seen from Table 3 that only one of the curve length errors is 1.7%, while the others are less than 1%. This amount of difference in curve length is negligible in human vision. It means that the effect spots of the irregular 3D yarn can be placed at the proper position of the 3D yarn and the key-point-mapping algorithm can be used to imitate a pre-designed irregular 3D yarns in any given geometric structure of fabric.

### 5.2. Application Prospect and Future Research

Theoretically, the mapping technique can also be extended to braided fabric as long as the structure is given. It can also be predicted that this mapping technique can be extended to the application of other fancy yarns such as flake yarn. For the folded fancy yarn, the central curve of the constitute single yarns are different from the central curve of the folded yarn, the mapping process will be more complex and requires further study. For uneven fancy yarns such as snarl yarn or loop yarn, the effect spots are not just the slub effect, new way of modelling effected 3D yarn is needed as well.

Another problem may be encountered when the key-point-mapping technique is applied. Due to the irregular cross-sections along the yarn central curve, two different yarns may collide at the thicker spots, which leads to an unreasonable 3D model of the geometric structure. The collision of the yarns should be detected and rectified in the future research.

## 6. Conclusions

This paper proposed a key-point-mapping algorithm to imitate pre-designed uneven 3D yarns in various geometric structures of fabrics. The premise for such algorithm is that the target curves of all the constituent yarns of the fabric structure are known in advance. The core of the mapping is to keep the curve lengths between the effect spots and the contour of the cross-sections corresponding to the effect spots of irregular/uneven 3D yarns unchanged whatever the central curve of the yarn changes. From the examples in this paper, the effect spots were well kept and reasonably distributed in the woven structure and knitted structure, which indicates that the key-point-mapping technique is reasonable, effective and accurate to apply pre-designed irregular/uneven 3D yarns in any fabric structure. The error analysis proves the conclusions further.

## Figures and Tables

**Figure 1 polymers-14-03992-f001:**
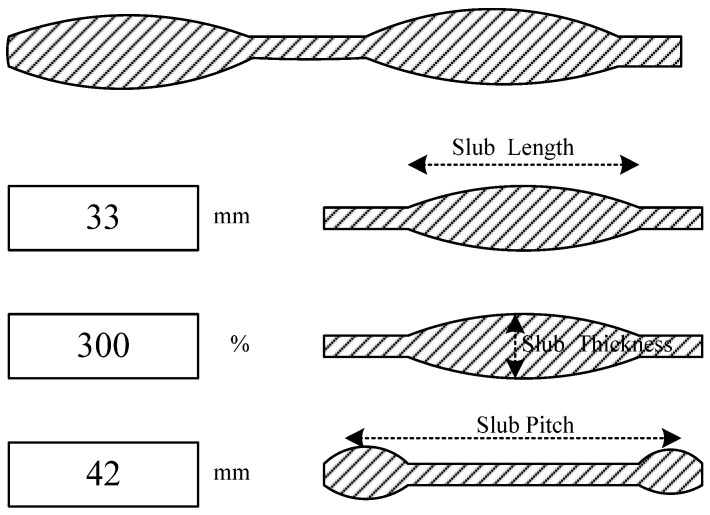
Three parameters to control the contour of slub knots.

**Figure 2 polymers-14-03992-f002:**
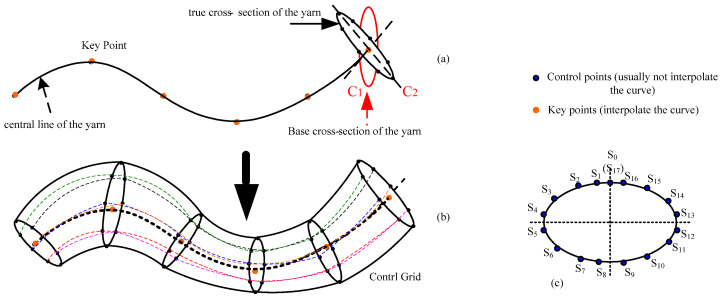
Principle of designing a 3D slub yarn. (**a**) Schematic diagram of the central curve and base cross-section of a 3D yarn. (**b**) The control grid for a 3D yarn modelled by B-spline curve. (**c**) Key control points for the cross-section of a 3D yarn.

**Figure 3 polymers-14-03992-f003:**
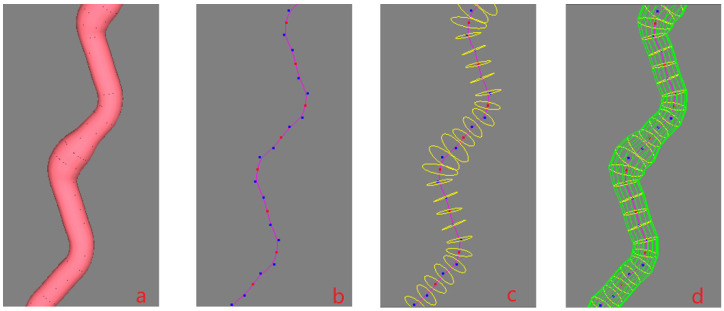
Design process of a 3D irregular yarn. (**a**) A solid 3D irregular yarn. (**b**) The control points of the cubic quasi-uniform B spline curve (red points: key points and control points, blue points: control points). (**c**) Cross-sections of the yarn. (**d**) Control grid of the B-spline surface for the 3D yarn.

**Figure 4 polymers-14-03992-f004:**
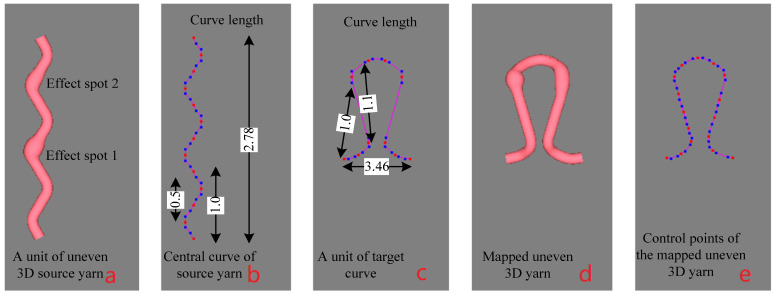
The procedure of changing the central line of a 3D slub yarn.

**Figure 5 polymers-14-03992-f005:**
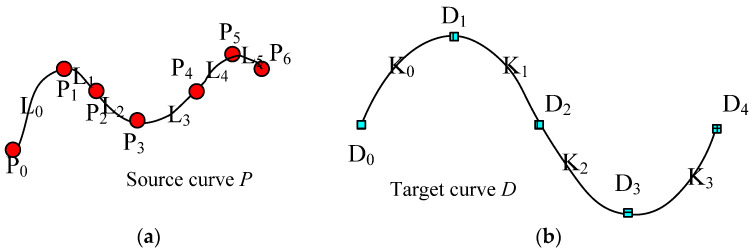
(**a**) Key points on the central curve *P* of source yarn. (**b**). Key points on the target curve *D*.

**Figure 6 polymers-14-03992-f006:**
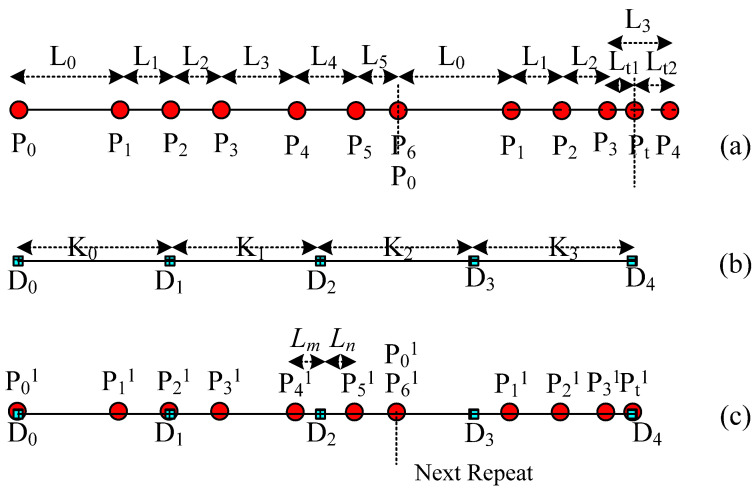
The calculation of key point spacing along the central curve and the combination of the sorting of key points. (**a**) Calculate the distance to the mapping start P_0_ for each key control point on the source yarn. (**b**) Calculate the distance to the mapping start D_0_ for each key point on the target curve. (**c**) Combine and rearrange all the key points of both source yarn and target curve in order on the new central curve according to the distances to the random mapping start *P*_0_ or *D*_0_.

**Figure 7 polymers-14-03992-f007:**
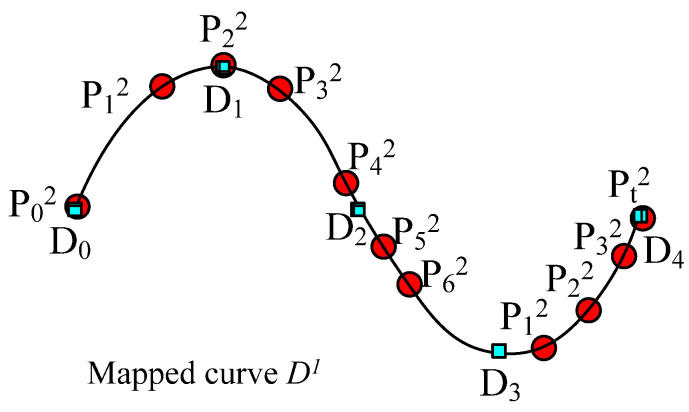
Arrangement of all key points in curve *D*^1^ according to distance.

**Figure 8 polymers-14-03992-f008:**
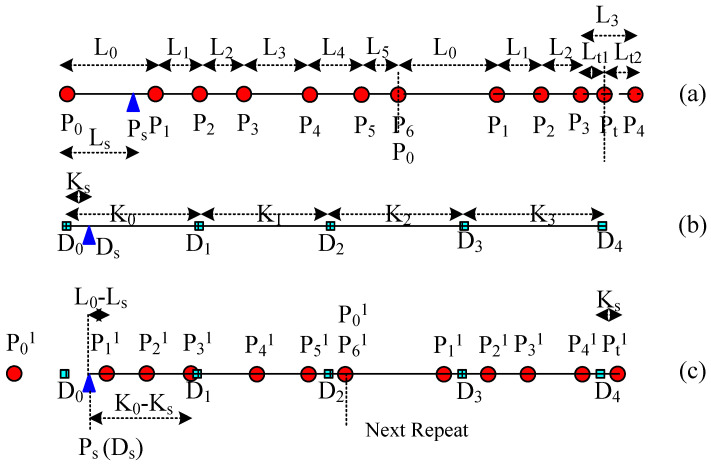
The mapping procedure for the different starting points. (**a**) Calculate the distance to the random mapping start *P_s_* for each key control point on the source yarn. (**b**) Calculate the distance to the random mapping start *D_s_* for each key point on the target curve. (**c**) Combine and rearrange all the key points of both source yarn and target curve in order on the new central curve according to the distances to the random mapping start *P_s_* or *D_s_*.

**Figure 9 polymers-14-03992-f009:**
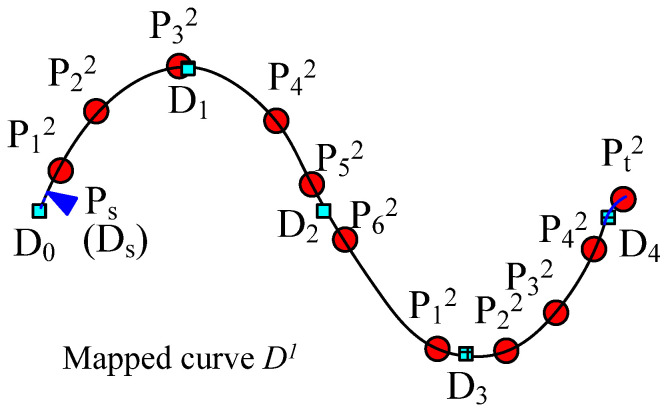
The mapping result for the different starting points.

**Figure 10 polymers-14-03992-f010:**
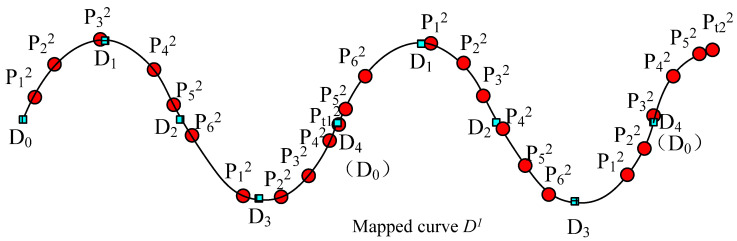
The mapping result for the multiple repeats of the curve.

**Figure 11 polymers-14-03992-f011:**
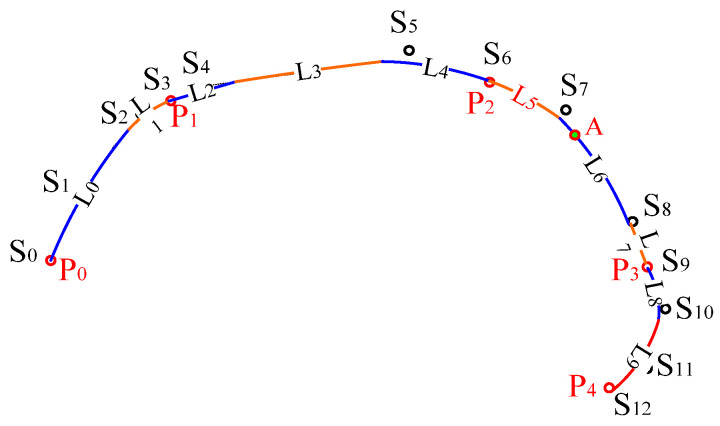
Key points, control points and curve segments of a shape-preserving quasi-uniform B-spline curve.

**Figure 12 polymers-14-03992-f012:**
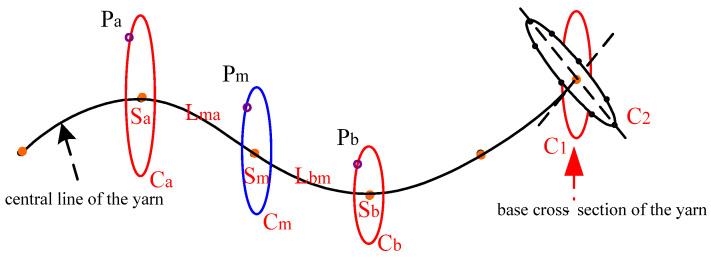
The method for generating a base cross-section by two known ones.

**Figure 13 polymers-14-03992-f013:**
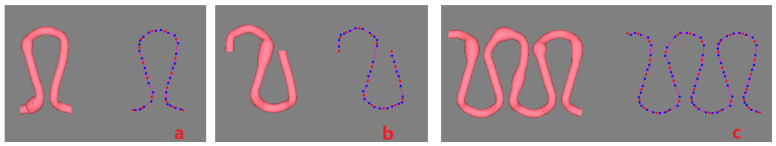
The mapping effect with different mapping starting points and mapping length. (**a**) Mapping a full unit length of the source yarn at the origin of the target curve. (**b**) Mapping a full unit length from the origin of the source yarn onto the target curve. (**c**) Mapping a random length of a given position of the source yarn onto the target curve at another specified position.

**Figure 14 polymers-14-03992-f014:**
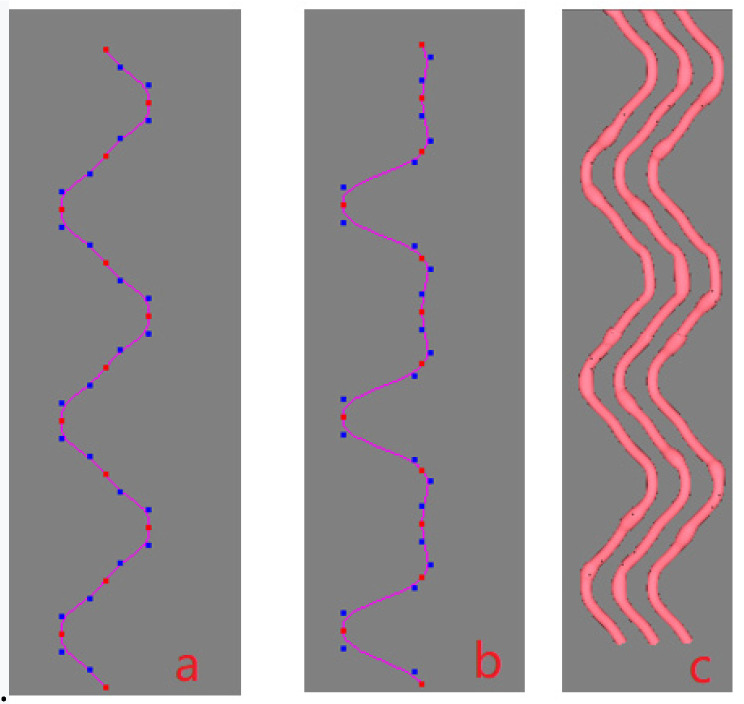
Different target curves and different mapping starts. (**a**) Target curve a. (**b**) Target curve b. (**c**) Mapping effects onto the target curve a from the source yarn at different mapping starts.

**Figure 15 polymers-14-03992-f015:**
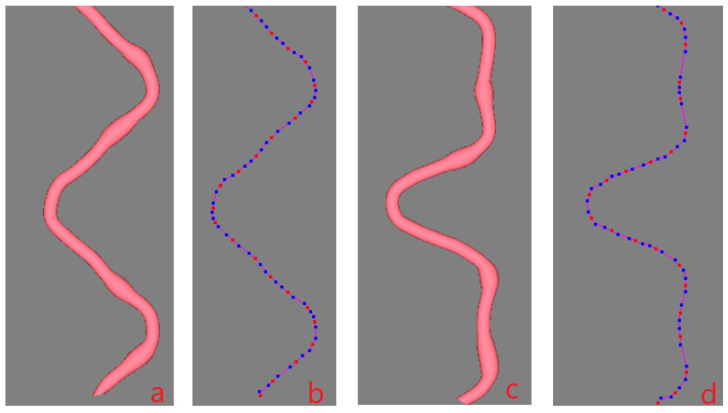
Enlarged generated central curves after mapping (local image). (**a**) The enlarged mapping effect of Figure 14c. (**b**) The control points of the mapped curve in Figure 15a. (**c**) The enlarged mapping effect onto the targe curve Figure 14b. (**d**) The control points of the mapped curve in Figure 15c.

**Figure 16 polymers-14-03992-f016:**
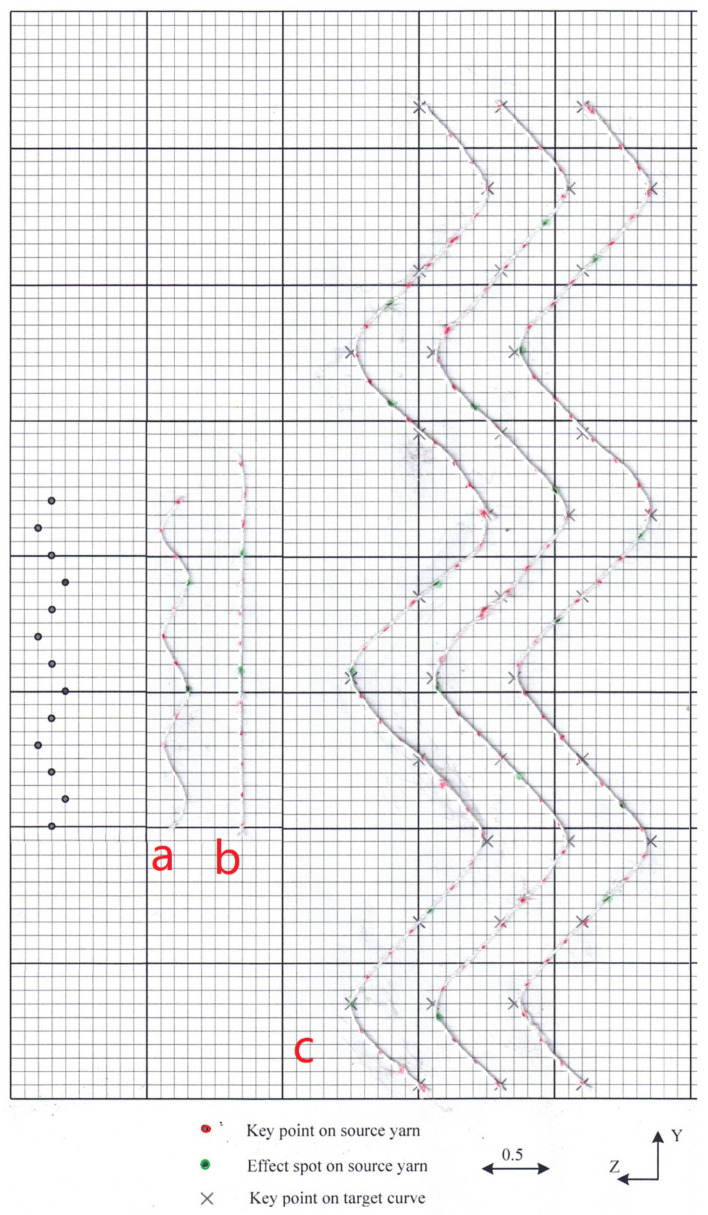
Real effect yarn bending diagram as compared with Figure 14c. (**a**) A full repeat unit of the real bent source yarn corresponding to Section A.1. (**b**) A full repeat unit of the straightened source yarn. (**c**) The mapping effect of the real yarns corresponding to Figure 14c.

**Figure 17 polymers-14-03992-f017:**
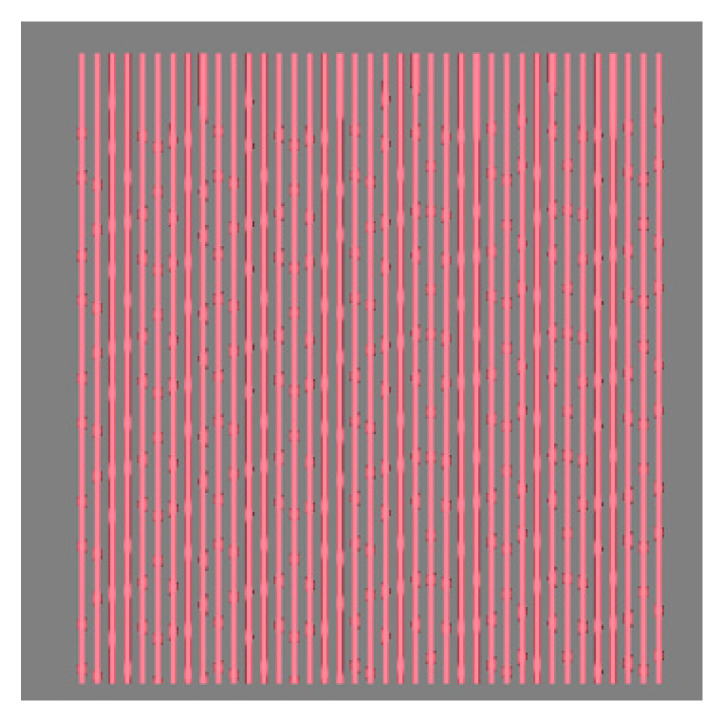
The imitated electronic blackboard.

**Figure 18 polymers-14-03992-f018:**
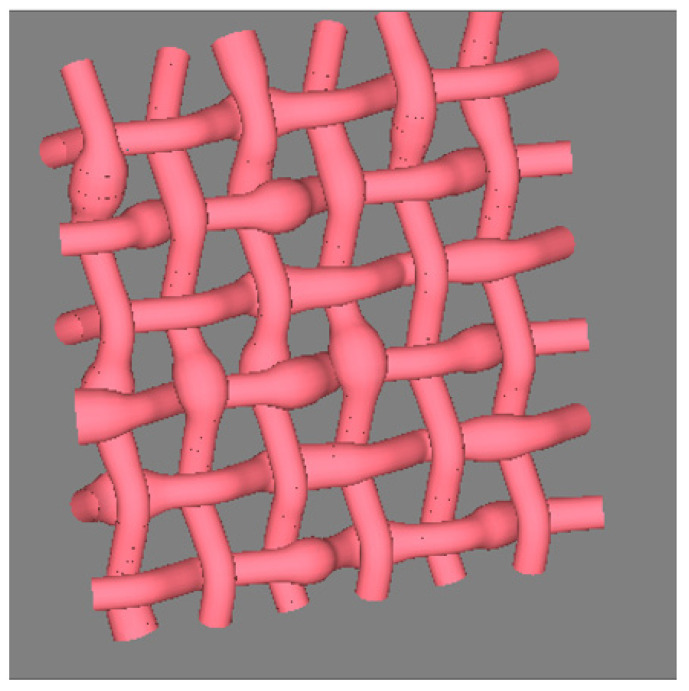
Woven structure with 3D slub yarns.

**Figure 19 polymers-14-03992-f019:**
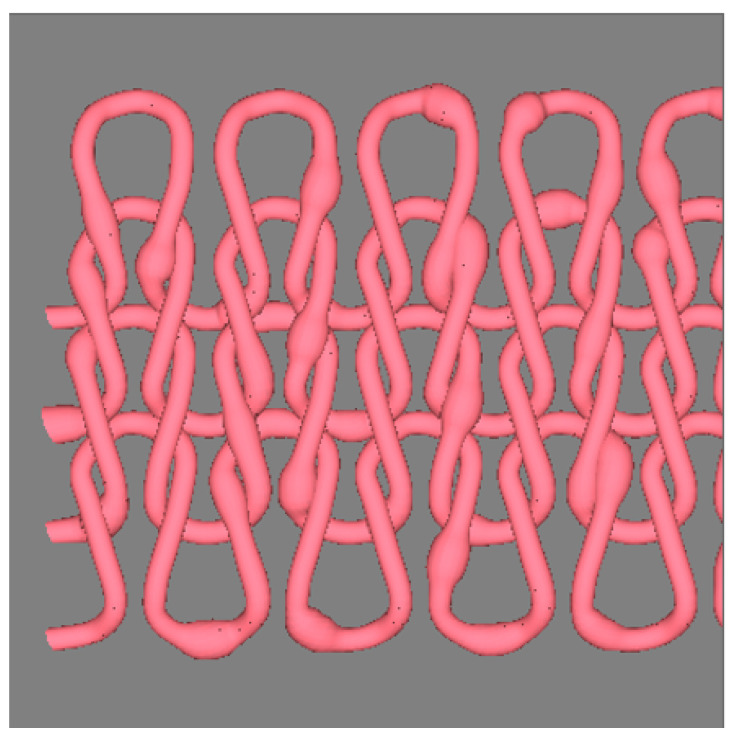
Weft-knitted structure with 3D slub yarns.

**Table 1 polymers-14-03992-t001:** Specifications of the source yarn and target curve.

Attribute	Data	Number of Key Points	Curve Shape	Length of the Curve Repeat	Positions of the Effect Spots (Indexed from 0)
Source yarn	Section A.1	13	Figure 4b	2.768905	No 5, No 9
Target curve	Section A.2	10	Figure 4c	3.460019	None

**Table 2 polymers-14-03992-t002:** Mapping parameters of the examples.

Mapping Example No	Mapping Starting Point of Source Yarn		Mapping Starting Point of Target Curve		Mapping Length	Figures of Mapping Effects
	Index Number of the Anterior Key Point	Length after the Anterior Key Point	Index Number of the Anterior Key Point	Length after the Anterior Key Point		
1	0	0	0	0	3.46	Figure 4d,e
2	0	1	0	0	3.46	Figure 13a
3	0	0	0	1	3.46	Figure 13b
4	1	0.5	2	1.1	8.8	Figure 13c

**Table 3 polymers-14-03992-t003:** The accumulative length errors at different mapping lengths.

No. of Target Curve	Target Curve of Single Repeat		Theoretic Mapping Length	Curve after Mapping		Accumulative Error (%)
	Number of the Key Points	Curve Length Repeat		Number of the Key Points	Actual Mapping Length	
1	10	3.4600189	8	36	8.137872	1.7234
1	10	3.4600189	12	53	11.941260	−0.4895
1	10	3.4600189	24	102	23.884195	−0.482521
1	10	3.4600189	28	123	27.893890	−0.378964
2	13	9.7456284	12	53	12.020981	0.1748417
2	13	9.7456284	28	120	27.884199	−0.413575
2	13	9.7456284	48	206	47.952755	−0.098427
2	13	9.7456284	52	223	51.793423	−0.397263
3	13	10.521817	12	50	11.905778	−0.785183
3	13	10.521817	28	114	27.793455	−0.737661
3	13	10.521817	48	195	47.566936	−0.902217
3	13	10.521817	52	211	51.721928	−0.534754

## Data Availability

Not applicable.

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
