# Peer review of "Imitation of a Pre-Designed Irregular 3D Yarn in Given Fabric Structures"

_polymers, 2022, doi:10.3390/polym14193992_

Round 1

Reviewer 1 Report

This research article describes the development of irregularity in the 3D yarn for the prediction of irregular structure in the fabric. A key point mapping algorithm between source yarn and target curve.

However, there are following concerns in this article.

1.       Introduction part needs to be modified. Authors should first explain the background and importance of this work.

2.       Recent relative research along with their draw backs should be added and research gap and its contribution should be added.

3.       Please update the overall references and include most recent research work in Introduction part.

4.       Reference techniques used in methodology such as Open GL. Similarly, line 91, “To solve the problem, a robust algorithm was proposed in Zheng [7] that inserting additional control points would ensure the curve to go through the key points.” Please explain in detail about Zheng work before using in your work.

5.       Line 135, “there is an assumption that no elongation and flatness happen when the yarn changes its path as the paper purely deals with a geometrical model rather than a physical model of fabric structure. How can we assume no elongation and compare geomatical model with physical samples?

6.       Is there any impact of number of control points? in key point modelling. Accumulative error is linked to key points?

7.       What are the limitations of key point mapping algorithm. Please describe them in detail. How we can compare error bars with previous work.

Author Response

First of all, I wound like to express thanks to the reviewer. 

  1. Introduction part needs to be modified. Authors should first explain the background and importance of this work.

The background part has been modified carefully for the readers who are not familiar with the CAD system for fabrics. The importance of the research is strengthened as well.

  1. Recent relative research along with their draw backs should be added and research gap and its contribution should be added.

Other researches or commercial CAD systems are lack of designing uneven or irregular 3D yarns, let alone how to demonstrate the 3D yarns in fabric structures. The previous work by the authors could demonstrate irregular 3D yarns in woven fabrics by manually assigning the cross-sections of the 3D yarns in special structure. However, when the fabric structure changes, the method does not work. The purpose of the study is the follow-up study to improve the algorithm.

  1. Please update the overall references and include most recent research work in Introduction part.

No new references are updated because few reports about the modelling of the irregular 3D yarns in geometric structure of fabrics are searched. Although there are some papers focusing on the modelling of the 3D yarn of individual fibers, it is not relevant to this paper and can only be used for the appearance of the yarn since it is not practical to use the complex model to predict the physical properties of the textiles.

  1. Reference techniques used in methodology such as Open GL. Similarly, line 91, “To solve the problem, a robust algorithm was proposed in Zheng [7] that inserting additional control points would ensure the curve to go through the key points.” Please explain in detail about Zheng work before using in your work.

The basic principle to model the yarn is explained in background part from line 50 to 60 and in section 2.2, “Representation of 3D irregular yarns”, from line 90 to 127.

  1. Line 135, “there is an assumption that no elongation and flatness happen when the yarn changes its path as the paper purely deals with a geometrical model rather than a physical model of fabric structure. How can we assume no elongation and compare geomatical model with physical samples?

When the fabric structure changes, the yarns in the fabric may be stretched. However, if the elongation is required, more parameters have to be considered, which makes the imitation more complicate and even impossible. For example, with the extension of the yarn in its central line, the contour of the cross-sections will shrink accordingly. Therefore, more parameters such as Young’s modulus, Poisson ratio, stress are required. In most cases for imitating the irregular yarns in fabric, the designers are more interested in the distribution of the effect spots of the yarn in fabric appearance. For the simplicity, no elongation is assumed in the study. If the elongation has to be taken into consideration, the algorithm can be improved by accordingly decreasing the curve length of the source yarn in section 3.4,” Locating of the corresponding point on the given SPQUCBSC”.

  1. Is there any impact of number of control points? in key point modelling. Accumulative error is linked to key points?

There are two kinds of control points, for the cross-sections or the central lines of the 3D yarn. The number of control points for each cross-section of the 3D yarn is fixed, i.e., 18. If the number is changed, the algorithm to calculate the contour of the cross-section will change as well. For the central lines of the 3D yarns, the number of the control points are provided by the fabric structure. The more the number of the control points, the more accuracy of calculation of the curved yarn length, the less the accumulative error occurs.

  1. What are the limitations of key point mapping algorithm. Please describe them in detail. How we can compare error bars with previous work.

To precisely demonstrate the proper position of the effect spots of the irregular 3D yarn in an arbitrary fabric structure, the number of the control points for the yarn path are increased 2 times at least, which requires a higher performance computer. It is almost impossible to demonstrate a 3D slub yarn with 3D twist effect in fabric structure if the yarn is relatively long. Therefore, there is no 3D twill effect shown in the imitated images in the manuscript.

Since there was no report for imitating irregular 3D yarns in an arbitrary fabric structure, the error in the manuscript are only compared between the fabric structure before and after mapping.

Reviewer 2 Report

Dear authors,

The data provided for the article entitled "Imitation of a pre-designed irregular 3D yarn in given fabric structures" are interesting; however, I am offering some comments throughout the manuscript including novelty assurance, scientific clarity and tactfulness, and a lot of typo and spacing errors.

(1) No need to use affiliation indicator if all authors indicate same affiliation (for example: Tianyong Zheng1,*, Wenli Yue1 and Xiaojiao Wang1 should be Tianyong Zheng*, Wenli Yue and Xiaojiao Wang).

(2) The readers would like to grab all of your key findings after having a look at the abstract. But the recent abstract looks like a combination of some statements. The abstract must be improved.

(3) Single keyword should not be too long like Shape preserving quasi uniform cubic B-spline curve (SPQUCBSC) (line 23-24).

(4) The introduction part is unprofessional; there is a lack of consistency between lines and paragraphs, it really needs to be revised very carefully.

(5) The main gap of the work is totally absent in the Introduction. The last paragraph of the Introduction should provide information (only) about the science gap in the previous studies and what motivates you to do this review with the objective of the study.

(6) It is recommended to use “Figure” instead of “Fig.” in the text throughout the manuscript.

(7) Every symbol used in the equations is not explained. These must be expained in the texts.

(8) Why do you used (...) (line 344 & 346)?

(9) Is this model applicable for all fibres (natural and synthetic)?

(10) Authors claimed that 'The paper proposed a key-point-mapping algorithm to imitate pre-designed uneven 3D yarns in various fabric structures'. Authors should specify the fabric strudcture.

(11) There are lots of typos and grammatical errors observed throughout the manuscript. These must be corrected and revised.

(12) Authors are asked to mention some conclusive findings by retaining coherence.

(13) References should be according to the journal template.

Author Response

First of all, I would like to express my thanks to you.

(1) No need to use affiliation indicator if all authors indicate same affiliation (for example: Tianyong Zheng1,*, Wenli Yue1 and Xiaojiao Wang1 should be Tianyong Zheng*, Wenli Yue and Xiaojiao Wang).

The affiliation indicators have been corrected.

(2) The readers would like to grab all of your key findings after having a look at the abstract. But the recent abstract looks like a combination of some statements. The abstract must be improved.

The abstract has been modified and the information about fabric CAD has been inserted for better understanding of the key findings of the paper. Some detailed information is deleted.

(3) Single keyword should not be too long like Shape preserving quasi uniform cubic B-spline curve (SPQUCBSC) (line 23-24).

The long keyword is replaced by B-spline curve.

(4) The introduction part is unprofessional; there is a lack of consistency between lines and paragraphs, it really needs to be revised very carefully.

The introduction part has been completely modified. The basic information about fabric CAD has been inserted, so the importance of the study and the technique gap can be easily understood.

(5) The main gap of the work is totally absent in the Introduction. The last paragraph of the Introduction should provide information (only) about the science gap in the previous studies and what motivates you to do this review with the objective of the study.

The introduction part has been completely modified. The motivation of the research and the science gap are explained in the last paragraph of the introduction part.

(6) It is recommended to use “Figure” instead of “Fig.” in the text throughout the manuscript.

All the words “Fig” have been replaced by “Figure”.

(7) Every symbol used in the equations is not explained. These must be expained in the texts.

All the symbols used in the equations are checked and explained.

(8) Why do you used (...) (line 344 & 346)?

“…” in line 344 and 346 represents the ellipsis.

(9) Is this model applicable for all fibres (natural and synthetic)?

In this model, the 3D yarn is considered as an integrity assembles of fibres rather than the individual fibres. Meanwhile, the contour of the 3D yarn is designed by geometrical equations. Therefore, it doesn’t make any difference whether it is natural fibres or synthetic faibres.

(10) Authors claimed that 'The paper proposed a key-point-mapping algorithm to imitate pre-designed uneven 3D yarns in various fabric structures'. Authors should specify the fabric strudcture.

In the manuscript, the path of the yarn in fabric structure is specified by a series of control points, or a cubic B-spline curve. There is no deformation and elongation of the cross-sections of the irregular 3D yarn when the fabric structure changes.

(11) There are lots of typos and grammatical errors observed throughout the manuscript. These must be corrected and revised.

The typos and grammatical errors have been checked and corrected sentence by sentence.

(12) Authors are asked to mention some conclusive findings by retaining coherence.

Conclusive findings are emphasized in the conclusion part by removing some future research to the discussion part. According to the study in the paper, the key-point-mapping algorithm is the effective way to reflect the effective spots of the pre-designed irregular 3D yarns to the proper position in the arbitrary given geometric structure of fabric.

(13) References should be according to the journal template.

References have been carefully modified according to the journal template.

Round 2

Reviewer 1 Report

Authors have incorporate all the required changes, therefore, it is recommended to accept paper.